# Improving quality and patient safety in surgical care through standardisation and harmonisation of perioperative care (SAFEST project): A research protocol for a mixed methods study

Claudia Valli[1,2]*, Willemijn L. A. Schäfer[3,4], Joaquim Bañeres[1,2,5], Oliver Groene[6,7], Daniel Arnal-Velasco[8], Andreia Leite[9,10], Rosa Suñol[1,2,5], Marta Ballester[1,2,5], Marc Gibert Guilera[1,2], Cordula Wagner[3], Hiske Calsbeek[11], Yvette Emond[11], Anita J. Heideveld-Chevalking[12], Kaja Kristensen[6], Lilian Huibertina Davida van Tuyl[3], Kaja Põlluste[13], Cathy Weynants[14], Pascal Garel[15], Paulo Sousa[9], Peep Talving[16,17], David Marx[18], Adam Žaludek[18,19], Eva Romero[8], Anna Rodríguez[1,2], Carola Orrego[1,2,4], for the SAFEST consortium¶

1 Avedis Donabedian Research Institute, Barcelona, Spain, 2 Universidad Autónoma de Barcelona, Barcelona, Spain, 3 Netherlands Institute for Health Services Research (Nivel), Utrecht, The Netherlands, 4 Department of Surgery, Northwestern Quality Improvement, Research & Education in Surgery, Northwestern University, Chicago, IL, United States of America, 5 Network for Research on Chronicity, Primary Care, and Health Promotion (RICAPPS), Barcelona, Spain, 6 OptiMedis AG, Hamburg, Germany, 7 Faculty of Management, Economics and Society, University of Witten/Herdecke, Witten, Germany, 8 Spanish Anaesthesia and Reanimation Incident Reporting System (SENSAR), Alcorcon, Spain, 9 NOVA National School of Public Health, Public Health Research Center, Comprehensive Health Research Center, CHRC, NOVA University Lisbon, Lisbon, Portugal, 10 Department of Epidemiology, Instituto Nacional de Saúde Doutor Ricardo Jorge, Lisboa, Portugal, 11 Scientific Center for Quality of Healthcare (IQ healthcare), Radboud Institute for Health Sciences (RIHS), Radboud University Medical Center, Nijmegen, The Netherlands, 12 Department of Operating Theatres, Radboud University Medical Center, Nijmegen, The Netherlands, 13 Department of Internal Medicine, Institute of Clinical Medicine, University of Tartu, Tartu, Estonia, 14 European Society of Anaesthesiology and Intensive Care (ESAIC), Brussels, Belgium, 15 European Hospital and Healthcare Federation, Brussels, Belgium, 16 Department of Surgery, Institute of Clinical Medicine, University of Tartu, Tartu, Estonia, 17 Department of Surgery, North Estonia Medical Centre, Tallinn, Estonia, 18 Spojená Akreditační Komise–Czech accreditation commission, Prague, Czech Republic, 19 Department of Public Health, Charles University, Third Faculty of Medicine, Prague, Czech Republic

¶ The complete membership of the author group can be found in the Acknowledgments.
* cvalli@fadq.org

## Abstract

### Introduction

Adverse events in health care affect 8% to 12% of patients admitted to hospitals in the European Union (EU), with surgical adverse events being the most common types reported.

### Aim

SAFEST project aims to enhance perioperative care quality and patient safety by establishing and implementing widely supported evidence-based perioperative patient safety practices to reduce surgical adverse events.

**Data Availability Statement:** The SAFEST project main data dictionaries and databases generated will be available on requests for uses related to research and quality improvement, and potentially for commercial exploitation, subject to approval by the consortium. Data availability and access is governed by the SAFEST Data Management Plan which is aligned with the EU Open Data Initiative and the FAIR Principles (34). Further details and information on how to access the data will be available from SAFEST project website (www. safestsurgery.eu).

**Funding:** This work was supported by the European Union under the Horizon Europe Research and Innovation Programme under the grant agreement n° 101057825. The funder had no role in study design, data collection and analysis, decision to publish, or preparation of the manuscript.

**Competing interests:** The authors have declared that no competing interests exist.

**Abbreviations:** CPGs, Clinical Practice Guidelines; EU, European Union; EUR, Euro; IRLM, Implementation Research Logic Mode; QILC, Quality Improvement Learning Collaborative; PROMs, Patients' reported outcomes; PREMs, Patient reported experience measur.

## Methods

We will conduct a mixed-methods hybrid type III implementation study supporting the development and adoption of evidence-based practices through a Quality Improvement Learning Collaborative (QILC) in co-creation with stakeholders. The project will be conducted in 10 hospitals and related healthcare facilities of 5 European countries. We will assess the level of adherence to the standardised practices, as well as surgical complications incidence, patient-reported outcomes, contextual factors influencing the implementation of the patient safety practices, and sustainability. The project will consist of six components: 1) Development of patient safety standardised practices in perioperative care; 2) Guided self-evaluation of the standardised practices; 3) Identification of priorities and actions plans; 4) Implementation of a QILC strategy; 5) Evaluation of the strategy effectiveness; 6) Patient empowerment for patient safety. Sustainability of the project will be ensured by systematic assessment of sustainability factors and business plans. Towards the end of the project, a call for participation will be launched to allow other hospitals to conduct the self-evaluation of the standardized practices.

## Discussion

The SAFEST project will promote patient safety standardized practices in the continuum of care for adult patients undergoing surgery. This project will result in a broad implementation of evidence-based practices for perioperative care, spanning from the care provided before hospital admission to post-operative recovery at home or outpatient facilities. Different implementation challenges will be faced in the application of the evidence-based practices, which will be mitigated by developing context-specific implementation strategies. Results will be disseminated in peer-reviewed publications and will be available in an online platform.

## Background

Adverse events in health care affect between 8% and 12% of patients admitted to hospitals in the European Union (EU) [1] and are accountable for approximately 21 billion Euros in direct costs [2]. An adverse event can be defined as 'an unintended injury or complication caused by healthcare management and not by the patients' underlying disease' [3]; adverse events cause potentially preventable patient harm, lengthen hospital stays, and increase healthcare costs [3]. Surgical-related adverse events are one of the most common adverse events reported [4], with surgery and intensive care showing the highest pooled prevalence of preventable patient harm [5]. A more recent retrospective cohort study showed that in a random sample of 2809 hospital admissions, 23,6% suffered of at least one adverse event of which 30,4% were related to surgical or other procedural events [6].

Adopting evidence-based practices can enhance the quality of care delivered to the patients and thus significantly improve the overall safety outcomes of surgical care [7; 8]. A retrospective cohort study including 25,513 adult patients undergoing non-day care surgery in a tertiary university hospital in The Netherlands, concluded that guideline adherence to evidence-based practices was strongly associated with a reduced mortality rate [9]. However, translating evidence into practice is slow in the healthcare field; in fact, research indicates that it can take up

to 17 years for evidence-based practices to be implemented, if implemented at all [10]. Common barriers to the implementation of evidence-based practices in clinical practice include: 1) front-line professionals do not know or do not trust the intervention effectiveness 2) evidence-based practices have been designed or implemented without consideration of patients' needs and preferences 3) implementation context is not considered 4) non-effective implementation strategies or 5) health-professionals do not have the appropriate support from managers and decision-makers [11]. Therefore, having a set of patient-centred, evidence-based, practices with a high level of consensus and a comprehensive tailored implementation strategy considering contextual factors at macro-, meso- and micro-levels, is a cornerstone to ensure access to safe, innovative, sustainable, and high-quality surgical care [12].

Several studies have shown that most surgical errors (between 53 to 70%) occur outside the operating room, before and after surgery [3]. Thus, to optimise surgical safety, there is a need for a shift towards an approach focusing on the entire surgical pathway, including pre- and post-surgery [13]. This also includes care provided outside the hospital, for example optimisation of co-morbidities management in the primary care setting before surgery and pain medications prescribed after surgery.

Following the above-mentioned, multidisciplinary teams—including public health researchers, experts on patient safety and perioperative care, evaluators, clinicians, economic analysts, scientists, patients' representatives, and policy advisors from seven European countries—developed the SAFEST project (https://www.safestsurgery.eu/).

## Objectives

The SAFEST project, that will be conducted from June 2022 to June 2026, aims to improve and harmonise perioperative quality of care and patient safety by reducing surgical adverse events through the establishment and implementation of widely supported perioperative patient-centred standardised practices.

We also aim to increase the adherence to patients' safety practices by identifying contextual factors (at macro-, meso- and micro-levels) that inhibit or promote the adoption, implementation, and sustainment of evidence-based patient safety practices. This information may be used to develop recommendations on the implementation of the standards in hospitals in various contexts across Europe to reduce knowledge-practice gaps.

## Methods

### Setting

SAFEST project will focus on the adult perioperative journey, including the following phases: pre-operative outside the hospital; pre-operative in the hospital; intra-operative in the hospital; post-operative in the hospital and post-operative outside the hospital (**Fig 1**).

### Design

The SAFEST study is a mixed-methods hybrid type III implementation study [14] supporting the development and implementation of evidence-based patient safety practices through a Quality Improvement Learning Collaborative (QILC) strategy and combining qualitative and quantitative evidence through a convergent segregated approach [15]. The QILC will adopt a participatory design involving multidisciplinary teams of patients and clinical care providers per participating hospital and related healthcare facility, including surgeons, anaesthesiologists, nurses, quality experts, IT representatives and primary care providers among others. The

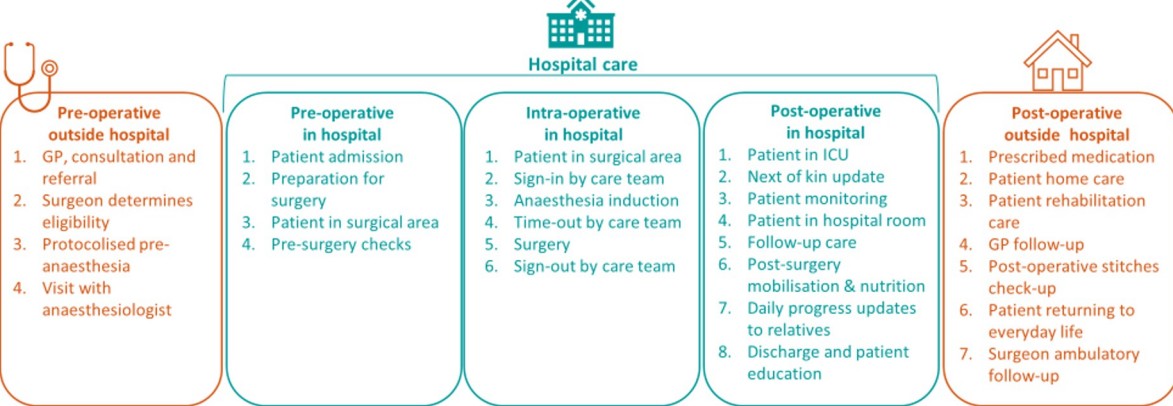

**Fig 1. Perioperative patient journey.** The project will be carried out in 10 hospitals and related healthcare facilities from 5 European Countries (Czech Republic, Estonia, Portugal, Spain, and The Netherlands). At the end of the project, a call for participation will be open to any interested hospitals in the EU and beyond.

QILC teams from various hospitals will gather for learning sessions to share and adopt best practices towards a common improvement goal.

## Guiding frameworks

The overall study design will be underpinned by the Implementation Research Logic Model (IRLM), which will support the implementation of the project by ensuring scientific rigour, reproducibility and testable causal pathways [16]. **Fig 2** shows how the IRLM will be applied within the SAFEST project and their elements. IRLM identifies a set of determinants from different domains of the updated Consolidated Framework for Implementation Research (CFIR) that can impact the implementation process, the mechanisms, and outcomes [17].

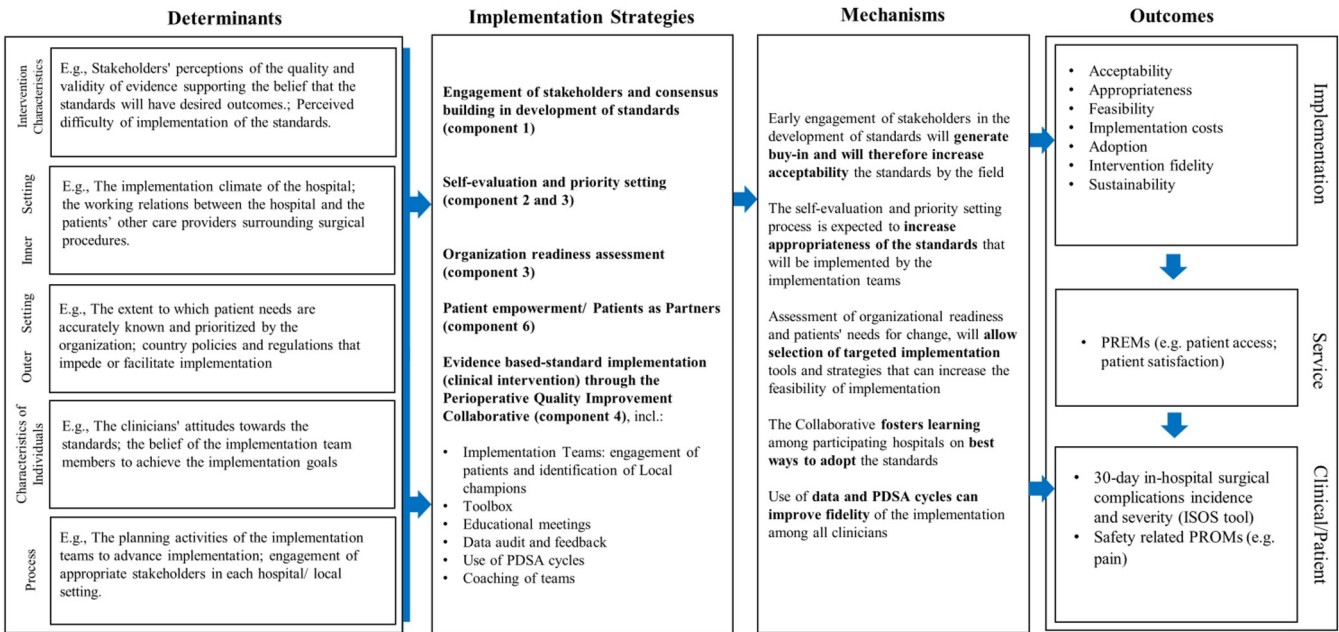

**Fig 2. IRLM applied within the SAFEST project.**

Additionally, an integrated sustainability framework will be developed by expanding the CFIR [17] to include sustainability determinants to guide the sustainable implementation of project intervention.

## Evidence-based patient safety practices

The evidence-based patient safety practices in SAFEST will include structural and process practices to promote patient safety and prevent the occurrence of adverse events and complications to be implemented throughout the continuum of perioperative care. The practices will be based on the highest level of available evidence indicating a positive impact on patient safety outcomes. These practices will be selected with patients and with relevant stakeholders including clinicians, hospital management, and national level policy makers.

## Implementation strategies

The Quality Improvement Learning Collaborative (QILC)–will serve as the main implementation strategy of the standardised practices. Within the QILC multidisciplinary teams from multiple hospitals come together in learning sessions to share and learn best practices to make improvements around a common goal. Based on previous studies [18; 19], we hypothesise that by implementing the standardised practices within a QILC, adherence to standardised patient safety practices in perioperative care will improve by 15% and the frequency of surgical complications will decrease by 8% after 18 months of its implementation. The QILC will promote learning exchanges on improvement efforts and care provider teams will be guided to conduct rapid-cycle quality improvement projects. Teams will meet quarterly for didactive sessions on implementation strategies and to exchange lessons learnt. QILCs have shown to produce significant improvements in targeted clinical processes and outcomes, including patient safety [20].

## Ethics

SAFEST project coordinator (Avedis Donabedian Research Institute) applied for ethical approval for the overall project to the local Clinical Research Ethics Committee (Comitè Ètic d'Investigació Clínica de l'IDIAP Jordi Gol) and it was granted on the 26th of July 2022 (22/146-P). In addition, each participating hospital from the five countries applied and obtain the ethical approval from their local authorities: from the Ethics Committee of the Centro Hospitalar e Universitario de Lisboa Central (1321/2022) and the Ethics Committee of the Centro Hospitalar e Universitário São João (20/2023) for Portugal; from the Clinical Research Ethics Committee (CEIM) of the Hospital Universitario y Politécnico La Fe (2022-815-1) and the Hospital Universitario Fundacion Alcorcon (22/107) for Spain; from the Ethics Committee of the Central Military Hospital Prague (108/18-45/2023) and the Opinion of the Ethics Committee on Clinical trial on Human Medicinal Products of Hořovice Hospital (066–2023) for Czech Republic; from the Human Research Ethics Committee of the University of Tartu for both Estonian hospitals: Tartu Hospital and of North Estonia Hospital (373/T-8); and, from the Medical Ethical Committee for Martini Hospital (2022–103) in the Netherlands. The Radboudumc university medical center in The Netherlands did not apply for the ethical approval as according to the local Research Ethics Committee, a full ethical review was not required (2022–15897).

## Outcomes

Underpinned by the Donabedian framework to evaluate quality in healthcare [21], we will include measures to evaluate patient safety practices adherence, clinical effectiveness, patients' perspectives, and the implementation process:

## PRIMARY OUTCOME

Research question: How effectively are patient safety practices being adhered to in perioperative care?

- Patient safety practices adherence will be assessed using data from electronic health records. These practices will be selected based on insights that will be gathered from self-evaluations and meetings with participating hospitals, focusing on areas with significant potential for improvement."

## SECONDARY OUTCOMES MEASURES

Research question: What is the impact of adherence to patient safety practices on clinical effectiveness?

- Clinical effectiveness will include 30-day in-hospital surgical complications incidence and severity—following elective inpatient surgery—using the ISOS tool developed by the International Surgical Outcomes.

  Research question: How do patients perceive the safety and quality of their surgical care?

- Patients' perspectives will include Patient Reported Measures (PROMs) and Patient Reported Experience Measures (PREMs).

  Research question: What are the key implementation process outcomes and how do these factors influence the successful implementation of standardized practices in surgical settings?

- The implementation process outcomes will include variables as acceptability, appropriateness, feasibility, implementation costs, adoptions, intervention fidelity and sustainability [22].

## Project procedures

The project will consist of 6 main components: 1) Development of the patient safety standardised practices, 2) Guided self-evaluation of the SAFEST standardised practices in ten hospitals across five countries and beyond, 3) Identification of priorities and actions plans, 4) Setting up of the Quality Improvement Learning Collaborative (QILC), 5) Evaluation of the strategy effectiveness and monitoring for patient safety improvement and 6) Patient empowerment for patient safety. **Fig 3** represents SAFEST overall approach and project's results.

## 1. Development of the patient safety standardised practices

The objective of this component is to develop and reach consensus on the set of evidence-based standardised practices. It will consist of five steps:

**1.1 Development of the initial list of standardised practices.**   We will conduct a systematic review to identify clinical practice guidelines (CPGs) for safety in the perioperative care process in adult patients and classify the included recommendations according to the strength and level of evidence, and CPGs quality using the AGREE II tool [23]. A protocol specifying inclusion criteria, data extraction and data analysis methods has been developed and registered in PROSPERO [24].

**1.2 Recruitment of multidisciplinary expert groups.**   We will recruit two multidisciplinary expert groups through purposive sampling: 1) a Scientific Advisory Group (SAG) and, 2) a Scientific Executive Group (SEG) of experts. These two groups will be involved throughout the patient safety practices development to provide inputs and feedback based on their expertise.

**Fig 3. SAFEST overall approach and project's results.**

They will be composed by multidisciplinary experts and relevant stakeholders including, patient representatives, perioperative scientific societies representatives, patient safety experts, medical industry members, national and international healthcare authorities, other related stakeholders, and partners organisation.

**1.3 Delphi survey (two rounds) with multidisciplinary expert groups and patients.** We will then produce a first draft of existing standardised practices that will be evaluated by the SAG through a modified Delphi technique consisting of a two-round online survey. SAG members will rate a) the importance and b) the feasibility of implementation of each practice using a 9-points Likert scale (1 being the least and 9 being the most important).

**1.4 European surgical patient safety conference.** Based on the feedback from the two-round Delphi survey we will refine the list and then present it for discussion in an in-person conference with a selection of SEG and SAG members balanced terms of gender, country and area of expertise.

**1.5 Development of the final list of the standardised practices.** Through an iterative approach, we will refine the list of the SAFEST standardised practices considering the feedback from the guided self-evaluation and the quality improvement learning collaborative of the 10 participating hospitals in Czech Republic, Estonia, Portugal, Spain, and The Netherlands (after component 2).

## 2. Guided self-evaluation of the SAFEST standardised practices in ten hospitals across five countries and beyond

The guided self-evaluation has the objective of evaluating the degree of implementation of the SAFEST standardised practices in ten hospitals at the time of evaluation. It will consist of four steps:

**2.1 Development and testing of an online platform with a self-evaluation tool.** We will develop an online platform incorporating a tool that will contain a scoring system to assess the degree of implementation of the SAFEST standardised practices resulting from 1.4 European surgical patient safety conference. The tool will be tested and then translated into the five main languages in participating hospitals (Czech, Dutch, Estonian, Portuguese, and Spanish). The platform will also include a forum/help desk feature through which participating hospitals can ask for help/assistance with the self-evaluation tool. Additionally, the platform will embed other relevant sections to collect hospitals data to evaluate on the adherence to the standardised practices and patients related surgical complications see components 5.2 and 5.3 of the Project procedures).

**2.2 Development of educational and training material.** We will develop educational and training materials including e a handbook providing instructions on how to use the online tool to conduct the self-evaluation. We will organise online training activities and webinars in addition to an in-person webinar in one participating hospital to present and explain the tool to the participating hospitals.

**2.3 Hospital guided self-evaluation.** The hospitals will adopt the tool and conduct a self-evaluation. We will analyse the collected data and report individualised results back to each hospital using the online platform.

**2.4 Open survey (self-evaluation of standards) of EU-27 hospitals and beyond.** At the end of the project, we will launch a call for participation to extend the self-evaluation of the SAFEST standardised practices in more hospitals from the 27 EU Member States and potentially outside the EU. Incentive to participate is to receive individualised benchmark reports based on the collected data and access to the materials toolbox. The results of this survey will be used to evaluate a potential scale-up of the standards implementation across Europe.

## 3. Identification of priorities and actions plans

This component aims to develop action plans for implementation of the standardised patient safety practices in the ten participating hospitals. It will consist of four steps:

**3.1 Systematic review of reviews on interventions improving patient safety.** We will conduct an umbrella review of effective evidence-based interventions to improve patient safety in the perioperative care process (including pre- and post-operative outpatient care in adult patients). A protocol specifying inclusion criteria, data extraction and data analysis methods has been developed and registered in PROSPERO [25].

**3.2 Prioritisation process.** We will then conduct webinars and interviews with hospital representatives, to discuss and identify priority improvement areas at hospital and country level based on the results of the previous performed self-evaluation. These representatives will also form the teams responsible for actual implementation of the patient safety practices (see also component 4).

**3.3 Driver diagram and barrier analysis.** We will analyse the data from the interviews, using a thematic analysis and summarising the main results of the prioritisation processes at the European level. Based on the results, we will develop a driver diagram indicating which factors need to be influenced to address the selected improvement priorities at the hospital level. A barrier analysis will be performed to get a full picture of barriers that may hinder

implementation and factors that may facilitate implementation of the improvement priorities selected. Facilitators and barriers will be analysed and categorised using CFIR framework [17].

**3.4 Action plans and toolbox development.** We will develop and outline action plans at hospital's level. We will provide support, for example through meetings and/or coaching sessions, for hospitals in (further) developing action plans based on improvement priorities, driver diagrams and identified barriers and facilitators, including formulation of improvement aims, interventions, working mechanisms, implementation strategies and evaluations (improvement cycle). Finally, we will develop a toolbox, including prioritisation instruction materials, tools for driver diagrams, barrier questionnaire and reporting tools, action plan formats, list of effective interventions and summary of results of the prioritisation process. This toolbox will be updated and refined during the Quality Improvement Learning Collaborative (see below).

## 4. Setting up of the quality improvement learning collaborative (QILC)

The QILC–implementation strategy of the SAFEST study- will work closely with the previous component (Identification of priorities and actions plans) and it consist of four steps:

**4.1 Identify members of quality improvement learning collaborative teams in each country.** We will form multi-professional implementation teams including patient representatives, and hospital personnel for each of the participating hospitals (ten teams in total).

**4.2 Toolbox dissemination and refinement.** We will disseminate to the implementation teams the refined and updated toolbox that was developed under component 3.4, including educational materials and instructions on how to apply it through the online platform. Through an iterative approach we will refine the toolbox during the implementation based on newly identified needs by the hospital implementation teams, hospital staff and patients.

**4.3 Facilitate learning and exchange to promote SAFEST standardised practices implementation.** Based on the action planning, the assessment of organisational needs and barriers (component 3) and evaluation of patient needs (component 6), we will develop a training curriculum for the hospital implementation teams to support the implementation of SAFEST standardised practices. The curriculum will include a series of activities, namely: six quarterly meetings for implementation teams consisting of didactic sections (e.g., on leadership engagement) and exchanges of experiences, tools, tips, and tricks; identification of external experts who will serve as quality improvement coaches; and bimonthly coaching sessions for the hospital implementation teams.

**4.4 Contextual factors evaluation.** We will develop data collection materials to measure implementation determinants, strategies, implementation outcomes, including sustainability and costs, based on the Implementation Research Logic Model (IRLM) [16] using the different domains of the CFIR [17]. We will collect qualitative data from the hospital implementation teams through group interviews and quantitative data from the hospital implementation teams through an online survey. We will then conduct thematic analysis of the interview transcripts, and we will narratively synthesise survey's results. Data analyses will focus on how contextual factors serve as barriers and drivers to implementation and on the importance of the QILC in combination with other implementation strategies employed at the 10 hospitals. Finally, we will provide feedback to hospital implementation teams on their implementation process.

## 5. Evaluation of the strategy effectiveness and monitoring for patient safety improvement

As part of the evaluation strategy, we will assess clinical effectiveness and implementation of the SAFEST standardised practices. This component will consist of five steps:

**5.1 Development of a core outcomes set for patient safety in perioperative care.** We will conduct an umbrella review to identify outcomes utilised to assess patient safety in perioperative care. A protocol specifying inclusion criteria, data extraction and data analysis methods has been developed and registered in PROSPERO [26]. The outcomes identified in the literature will then be prioritised through a two-round Delphi and stakeholder consensus. The results of the consensus process will inform the final core outcomes set (COS) for patient safety in perioperative care. A protocol detailing the COS development has been registered in COMET [27].

**5.2 Definition and preparation of evaluation procedures and guidelines at hospital level.** We will develop an evidence map of existing data and format across participating hospitals as well as cost-related data. Following this assessment, we will conduct a systematic retrospective record review of the health records at the patient level from the participating hospitals to assess ***a) adherence to the standardised practices*** and ***b) surgical complications measured with the ISOS tool*** developed by the International Surgical Outcomes Study [28]. Health records review will take place between June 2023 and November 2025.

**5.3 Evaluate effectiveness of QILC intervention strategy.** We will design an interrupted time series to estimate effects of the intervention on the adherence to the SAFEST standardized practices and the occurrence of surgical complications. This data will be collected into the SAFEST online platform (see component 2.1 of the Project procedures).

**5.4 Monitor improvements through evaluation of locally collected process and clinical data, patients' reported outcomes (PROMs) and patient reported experience measure (PREMs).** Each hospital will also be able to collect process and clinical data at the hospital level into the SAFEST platform to calculate measures as required from the COS developed as part of step 5.1. The platform will facilitate its analysis and visualisation through which we can monitor the implementation and provide feedback to the hospitals.

We will also utilise validated PROMs and PREMs questionnaires from the literature. Preference will be given to measures that have already been validated in the participating countries. The PROMs and PREMs questionnaires will be uploaded into a mobile app and integrated into the SAFEST platform. PREMs will be collected only after discharge, while PROMs before and after surgery. Patients will be invited to participate and report their PROMs and PREMs through the app. Authors will not have access to patient's identity during or after the study.

**5.5 Final assessment including triangulation of quantitative and qualitative data.** We will triangulate quantitative and qualitative data to assess the impact of the SAFEST strategy through a convergent segregated approach [15], including comparison between hospitals. We will conduct a sequential analysis of the quantitative and qualitative components of the data. We will analyse each dataset separately and then, draw meta-inferences informed by the findings from both data sets.

## 6. Patient empowerment for patient safety

Throughout the entire span of the project, we aim to include the patient's perspective in every activity undertaken. To ensure this, the following three steps will be taken:

**6.1 Engagement of patient expert groups.** Patients will be included as patient safety partners throughout the project, including in the co-design of the standardised practices, selection of priorities at local level, and development of evaluation measures. We will recruit a patient-researcher as a consortium team member to pro-actively conduct research activities. Patients and the patient researcher will constitute an expert group to oversee project activities and to participate in standards development, the Delphi survey, and meetings. Further, the expert

group will review the main materials and deliverables, ensuring that patient perspectives, gender issues and other elements regarding inequalities are incorporated across the project. Within SAFEST, co-production is expected to support patient-centred implementation of the intervention, patient satisfaction, and innovation [29].

**6.2 Identification of patients' needs.** Prior to implementation of the standards, researchers at country-level will conduct qualitative interviews with 12 purposefully selected adult patients in each of the ten hospitals (up to 120 patients in total) to identify patient needs, expectations, and priorities at the hospital and country level. We will develop recruitment material and strategies ensuring broad representation, including vulnerable populations. Patients will be recruited between September 2023 and December 2023. Based on the interviews and literature we will develop a patient journey map on the perioperative process in the hospitals. The journey maps and descriptions of interview results will be used to select strategies and actions for the implementation toolbox.

**6.3 Develop recommendations and tools for patient empowerment for patient safety.** We will develop recommendations and tools for empowerment of patients in the field of patient safety together with the patient-researcher and patient expert group. These tools will be developed as a result of the implementation component and may include a variety of supporting material, e.g., information on questions they could ask during pre- and post-operative consultations, a lay language informational video and project summary in collaboration with patient expert groups.

## Sustainability

Targeted actions will be embedded within the implementation process to promote sustainment of project impact beyond the duration of the SAFEST study. We will develop an integrated sustainability framework to guide the sustainable implementation of the QILC. This will be achieved by first conducting a literature scan to identify sustainability theories, models, and frameworks; extracting and standardizing the set of sustainability constructs; compiling and mapping final sustainability constructs onto the CFIR. This updated framework will serve as a common point of reference to guide interviews with key experts on sustainability factors and, eventually, more sustainable implementation of quality improvement activities. We will further conduct a time-driven activity-based costing exercise for the QILC implementation at participating hospitals to estimate the resources and time required for routine implementation of the developed tools when scaling up to a larger number of hospitals. Activity-based costing is an innovative approach towards measuring costs across an entire care episode whereby costs are estimated using the quantity of time and the cost per unit of each resource used [30]. This bottom-up approach to costings follows the actual care processes and contrasts with traditional costing approaches where cost estimates may be inaccurate. We will conduct activity-based costing for 3 perioperative care pathways focused on in this project. Using the patient journey maps produced, we will identify common touchpoints with the healthcare systems and compare the pre- and post- intervention costs for the 3 perioperative pathways. Results of this costing exercise will inform the business case development where costs are presented alongside the evaluation of intervention effectiveness and patient reported outcomes. Additionally, the information acquired from the costing exercise it will be used in drawing up business plans in our strategy to scale up implementation. In terms of environmental sustainability, we will draw on existing standards developed in partnership with HealthCareWithoutHarm [31], which addresses generic action areas to improve environmental sustainability of hospitals. These generic standards will be supplemented by specific perioperative-care standards, aligned with the activities implemented as part of the bundle [32, 33].

## Discussion

### Contribution

The SAFEST project represents a significant advancement in the dissemination and implementation of patient safety standardized practices within the continuum of care for adult patients undergoing surgery. By focusing on this critical area of healthcare, our study emphasizes the importance of patient safety across the entire surgical journey, from preoperative preparation to postoperative care.

Moreover, the SAFEST project stands out for its commitment to patient-centered care through the involvement of patient representatives throughout the project. Our study ensures that the perspectives and preferences of those directly impacted by surgical care are central to our efforts by actively involving patient representatives throughout the project.

Furthermore, the SAFEST project's emphasis on context-specific implementation strategies underscores its contribution to advancing the field of implementation science. Through the identification of contextual factors at micro, macro, and meso levels within each hospital setting, we will tailor our implementation strategies to address the specific challenges and opportunities present in each context. This approach not only increases the likelihood of successful implementation but also enhances the sustainability and scalability of patient safety practices in surgical care settings.

In summary, by enhancing dissemination and implementation efforts for patient safety standardized practices in the continuum of care for adult surgical patients, the SAFEST project makes a significant contribution to improving healthcare quality and patient outcomes. Through its patient-centered approach, tailored implementation strategies, and engagement with diverse stakeholders, our study is well-positioned to drive meaningful advancements in patient safety and quality improvement initiatives.

### Strengths and limitations

#### Strengths.

- We will develop a list of patient-centred evidence-based standardised practices following an iterative mixed-methods Delphi consensus-building approach involving a wide range of stakeholders.

- This project will result in a broad European implementation of evidence-based practices for perioperative care, spanning from the care provided before hospital admission to post-operative recovery at home or outpatient facilities.

- We will develop an interactive platform to support the adoption of the evidence-based standardised practices. The platform will be open to any hospital interested in participating.

- The perspective of patients will be actively included in every step of the project.

- We will conduct a comprehensive analysis of barriers and facilitators associated with the implementation process and sustainability of implementation.

#### Limitations.

- There will be variability in interpreting how to implement and assess the level of adherence of the evidence-based practices; specific implementation and evaluation manuals will be developed to favour a similar understanding across participating hospitals.

- Various implementation challenges are anticipated in the application of the evidence-based practices, including potential resource constraints unique to each participating health facility. These challenges will be mitigated by developing context-specific implementation strategies.

- The study operates within a defined timeframe, limiting the ability to capture long-term outcomes and trends. Nevertheless, we've strategically allocated resources to ensure that the impact of our project extends well beyond the immediate study duration.

## Dissemination of the results

The project's results will be disseminated in a tailored multi-pronged approach, including the creation of an interactive platform. The data generated by the project will be managed following the Golden Open access to maximise the impacts of the project's research outputs.

The results will be disseminated and exploited based on a Dissemination, Exploitation and Communication Plan (DECP) elaborated by the SAFEST Consortium to ensure that the project results will create impact in the long run and that will be used by relevant stakeholders. The DECP will benefit the project by increasing the visibility of the research as well as enhance project's reputation and pave the way for further collaboration and policy change.

## Acknowledgments

We would like to thank all the members of the SAFEST Consortium for their valuable assistance in developing this protocol: María del Mar Fernandez, Marieke Voshaar, Pedro Casaca-Carvalho, Patrícia Sousa-Paulo, Ana Beatriz Nunes, Víctor Soria, Yolanda Sanduende, Neus Fábregas, Ismael Martínez, Irene León, Ashish Bartakke, Javier Silva García, Joel Starkopf, Janne Kommusaar, Janne Pühvel, Mari Kangasniemi, Sofía Carbonell, Marie Nabbe, Maria Wittmann, Arta Leci, Julia Dowel, Alex Rawlings, Raffaella Donadio, Sylvia Daamen, Frantisek Vlcek, Sandro Zamarian,.

We also would like to thank the health professionals from the 10 participating hospitals for their involvement: Marion van der Kolk, Jan Hofland, Sjoukje de Vet-Kersten (**Radboud university medical center**); Henriëtte Smid-Nanninga, Hans de Boer (**Martini Hospital**); Nuno Diogo; Maria José Maia (**Centro Hospitalar Universitário Lisboa Central (CHULC, EPE)**); Ana Azevedo, Sara Rodrigues, Elsa Guimarães, Ricardo São Simão, Fernanda Bastos (**Centro Hospitalar Universitário de São João (CHUSJ, EPE)),** Paula Pérez, Laura Bruno, Mónica Millan Scheiding, Mª Jose Felip, Marisa Correcher Palau, Celia Lucas Jiménez, Ángel Garay, Diego Moreno, Carmen Reina Diaz-Crespo, EmilioPérez, Carlos Rabadan Sanz, Susana Gómez Leiba (**Hospital Universitari i Politècnic La Fe**), Rodrígo Molina, Laura Vidaurreta, Cristina Garmendia, Antonio Bartolome, Laura Vega, Isabel Sastre, Susana Lorenzo, Ana Belén Díaz, Inés de la Fuente, Alicia González Davila, Covadonga del Pozo (**Hospital Universitario Fundación Alcorcón**), Liis Jaanimäe, Maret Laheveer, Ilona Pastarus, Juri Karjagin, Kaie Stroo, Joel Starkpof (**Tartu University Hospital**), Olav Tammik, Kristina Lillemets (North Estonia Medical Centre), Helena McMenamin, Lucie Kubátová, Michal Ondica, Lenka Gutová, Jan Frühauf, Michaela Kalinová (**Military University Hospital Prague**), Blanka Hošková, Martina Landová (**Hořovice Hospital**).

## Author Contributions

**Conceptualization:** Willemijn L. A. Schäfer, Joaquim Bañeres, Oliver Groene, Daniel Arnal-Velasco, Andreia Leite, Rosa Suñol, Cordula Wagner, Hiske Calsbeek, Kaja Kristensen,

Lilian Huibertina Davida van Tuyl, Cathy Weynants, Pascal Garel, Peep Talving, David Marx, Adam Žaludek, Eva Romero, Carola Orrego.

**Methodology:** Willemijn L. A. Schäfer, Joaquim Bañeres, Oliver Groene, Daniel Arnal-Velasco, Andreia Leite, Rosa Suñol, Hiske Calsbeek, Yvette Emond, Kaja Kristensen, Lilian Huibertina Davida van Tuyl, Kaja Põlluste, David Marx, Adam Žaludek, Carola Orrego.

**Writing – original draft:** Claudia Valli, Willemijn L. A. Schäfer, Joaquim Bañeres, Oliver Groene, Daniel Arnal-Velasco, Kaja Kristensen, Carola Orrego.

**Writing – review & editing:** Claudia Valli, Willemijn L. A. Schäfer, Joaquim Bañeres, Oliver Groene, Daniel Arnal-Velasco, Andreia Leite, Rosa Suñol, Marta Ballester, Marc Gibert Guilera, Cordula Wagner, Hiske Calsbeek, Yvette Emond, Anita J. Heideveld-Chevalking, Kaja Kristensen, Lilian Huibertina Davida van Tuyl, Kaja Põlluste, Cathy Weynants, Pascal Garel, Paulo Sousa, Peep Talving, David Marx, Adam Žaludek, Eva Romero, Anna Rodríguez, Carola Orrego.

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
