## [Decision Letter · Decision Letter 0]

27 Feb 2024

PONE-D-23-31626Improving quality and patient safety in surgical care through standardisation and harmonisation of perioperative care (SAFEST project): a research protocol for a mixed methods study.PLOS ONE

Dear Dr. Valli,

Thank you for submitting your manuscript to PLOS ONE. After careful consideration, we feel that it has merit but does not fully meet PLOS ONE’s publication criteria as it currently stands. Therefore, we invite you to submit a revised version of the manuscript that addresses the points raised during the review process.

As you can se there is a descripancy between the reviews (reviewer 1: minor revisions; reviewer 2: major revision). In order to meet PLOS ONE's required publication standard you need to revise the manuscript in concordance with their comments.

Looking forward seeing your revised manuscript.

We look forward to receiving your revised manuscript.

Kind regards,

Ann-Sofie Sundqvist, Ass. Prof.

Academic Editor

PLOS ONE

2. Thank you for stating the following financial disclosure:"This work was supported by the European Union under the Horizon Europe Research and Innovation Programme under the grant agreement nº 101057825."

3. Please expand the acronym “EU” (as indicated in your financial disclosure) so that it states the name of your funders in full.

Reviewers' comments:

Reviewer's Responses to Questions

**Comments to the Author**

1. Does the manuscript provide a valid rationale for the proposed study, with clearly identified and justified research questions?

Reviewer #1: Yes

Reviewer #2: Yes

2. Is the protocol technically sound and planned in a manner that will lead to a meaningful outcome and allow testing the stated hypotheses?

Reviewer #1: Yes

Reviewer #2: Partly

3. Is the methodology feasible and described in sufficient detail to allow the work to be replicable?

Reviewer #1: Yes

Reviewer #2: Yes

4. Have the authors described where all data underlying the findings will be made available when the study is complete?

Reviewer #1: Yes

Reviewer #2: Yes

5. Is the manuscript presented in an intelligible fashion and written in standard English?

Reviewer #1: Yes

Reviewer #2: Yes

6. Review Comments to the Author

You may also provide optional suggestions and comments to authors that they might find helpful in planning their study.

Reviewer #1: This protocol describes a mixed-methods hybrid type III implementation study designed to support the development and adoption of evidence-based practices in order to improve perioperative patient safety. The study will be conducted in 10 hospitals of 5 European countries. Implementation outcomes include 1)adherence to standardised practices, 2)sustainability 3) usability, and 4) validity. The study will also assess adverse outcomes following surgery.

The study protocol is well developed and the methodology adapted to the study purpose. The only point that should be clarified is the link between context analysis through the CIFER protocol and the specific intervention that will be implemented. The study protocol makes the assumptions that it will be a teaching program, but actually, the results of context analysis may suggest alternative strategies than teaching (i.e. computer program, change of the process of care, walk rounds etc…) This should be discussed in the protocol further. There is also an imprecision in the introduction with the citation of the study by “Van Klei WA, Hoff RG, Van Aarnhem EE, et al. Effects of the introduction of the WHO "Surgical Safety Checklist" on in-hospital mortality: a cohort study. Ann Surg 2012;255(1):44-9 , as a reference showing that guidelines decrease surgical mortality. The study by Aarnhem is about a checklist, not about a guideline. I would suggest another reference. The limitation section is a bit vague since obstacles of implementation studies are well known and deserve attention (i.e. ressources, training, duration ). This should be specified in the limitation section.

In conclusion, a nice protocol of a promising study that needs however additional details provided.

Reviewer #2: General comments to authors

Thank you for allowing me to review the protocol for this ambitious but important project funded by the EU – Horizon Europe Framework Programme.

Since this is a substantial project I understand that it is a challenge to describe this in a balanced way in a protocol. But the paper will benefit from being restructured and clarified which will make it easier for the reader to grasp the project, see comments below. I think the main problem with the structure is that you have three processes (phases) sometimes running in parallel, that is the intervention (with planning and developing), the implementation and the evaluation process, but I have written some suggestions for this.

Although guidelines do not solve the whole issue to create safe care, to me this is an important prerequisite with impact on other initiatives that has to be solved. From a context perspective the need is extensive as the surgical specialty is responsible for many patient injuries or adverse events. I am looking forward to see future results from this project.

Specific comments

Title

Improving quality and patient safety in surgical care through standardisation and

harmonisation of perioperative care (SAFEST project): a research protocol for a mixed

methods study.

The title includes both concepts patient safety and quality, but is not explained in the background why the reader get unsure how it is used in the different sections in the paper.

Abstract section

- Needs reformulation according to recommendations below.

Background

Page 3.

- The background is somewhat short but logic except from that some main concepts are missing (see further down) and there is no clear rationale (please add this).

- Is it quality of care or patient safety that is your main focus? Please clarify.

- Furthermore, your main concepts of quality of care or patient safety which also is your outcome measures are barely not mentioned nor defined in this section.

- In the first section important information about adverse events is described. However, there is no clear definition of one of your main concepts i.e. adverse event (AEs). I find it very important that you define and are clear with what AE and patient injuries means for this project.

Page 4.

- Your objective is to improve and harmonise perioperative patient safety, but what does this mean. The focus in the background section is on unsafe care, clarify what you mean by perioperative patient safety? What do you want to achieve, is it less numbers of AEs which is implied? What about outcomes on a patient level? This needs to be developed.

- Is it quality of care or patient safety or both that is your main focus? Please clarify.

- One of the most important measures to achieve patient safety (safe care) in complex systems is to adopt a system perspective. According to me you are adopting a system perspective (system safety) in this project, which is very good. But the paper would benefit from a clearer description of this standpoint.

- In the last section you write multidisciplinary teams – including experts on patient safety and perioperative care. However, this is the only place in your paper where they are mentioned. Since patient safety is your overarching objective, I find it very important that they will be included during the whole process.

- Another concern is that you need to clarify what professions these multidisciplinary teams will consist of?

Objectives

- To clarify this project, it would benefit from identifying overarching research questions. Since you have three main outcomes (clinical effectiveness, patient´s perspectives and the process of implementation) in focus, this would be feasible.

Methods

Page 4. This study is a mixed-methods hybrid type III implementation study, the paper and project would benefit from a short explanation/motivation of why and by adding a reference to this. For example, Landes et al. (2019) https://www.ncbi.nlm.nih.gov/pmc/articles/PMC6779135/

- You also write that you will use mixed methods to collect data, add reference to this, for example Creswell & Plano-Clark. I also recommend you describe which design (convergent parallel design or what?), see recommended reference.

- You describe both design and data collection under the design heading. For a clear structure I recommend you to use the sub-headings design, setting (and sample) and data collection.

- Page 4. Quality Improvement Learning Collaborative (QILC) needs to be described and clarified early in the paper. As it is now, this is described at page 5.

- Good that you use Implementation Research Logic Model (IRLM) and CFIR as it clarifies and add to the written text. The paper would benefit from a sub-heading that e.g. says “guiding frameworks”.

- You write; The evidence-based patient safety practices in SAFEST will include structural and process practices to promote patient safety and prevent the occurrence of adverse events and complications to be implemented throughout the continuum of perioperative care.

• Since you have clear project outcomes, and you are adding structure and process, the paper should benefit from adding the reference from your institute Donabedian, A. (2005), Evaluating the Quality of Medical Care. The Milbank Quarterly, 83: 691-729.

• Check this section for sighting error, can be misinterpreted.

• This section explains parts of my concerns described under the background section, please add and explain this in that section.

• Furthermore, above the concept of complications is used but not explained in the background section.

Page 5 (first section) you write that QILC will serve as the main implementation strategy of

the standardized practices. As it is now you mix both hypothesis and strategies, please structure it more clearly. Add a sub-heading for example implementation strategies, and it will become clearer. What other strategies will you use (facilitation, training and/or support) and if, what does it include (content, activities, dose and delivery)?

Setting

Page 5. Under this section you describe that it is the adult perioperative journey with its phases and in or outside the hospital. The paper would benefit from a short description to all readers of what pre, per and post actually means.

Page 5. You write that; the project will be implemented in 10 hospitals from 5 European Countries. But as you focus on the whole patient journey that is inside and outside the hospital there is a need for a further explanation.

Outcomes

Page 5. You write that you will use a composite indicator based on the main priorities established by the participating hospitals. Please explain this briefly.

Described variables to measure implementation outcomes are you really going to use all 7 variables by Proctor? I think intervention fidelity (with its potential moderators) would be enough to evaluate the primary outcome measure adherence to patient safety practices. Rethink and clarify this.

Project procedures

Component 1.

Page 6. The project will consist of 6 main components, please add the six components in the text as it would clarify for the reader. But I can also see that there are different overarching phases which may help the reader to get structure i.e. development/design and planning phase, implementation phase and evaluation and reporting phase. Maybe these headings could help the reader to easily grasp the overarching structure of the work process.

Page 6. Figure 3 is very good to give structure to this project. Although a hybrid type 3 design, has its main focus on quality of care or patient safety implementation outcomes, it is confusing that the part with the development of the clinical intervention is presented after the Implementation Research Logic Model (IRLM) in the paper. It would benefit from describing the connection between figure 1 and 3, that is figure 3 (SAFEST overall approach) is the same as the block clinical intervention in figure 1 and if you could present it in a more logic way.

Page 6 _1.2 You will recruit multidisciplinary expert groups 1) a Scientific Advisory Group

(SAG) and, 2) a Scientific Executive Group (SEG) of experts. Further down in the paper you describe the composition of these groups, but patient safety experts are not mentioned here which I find a bit strange. It does not say on what system level, but patient safety experts from my experience exist even on macro level. Please clarify.

Furthermore component 1 seems clear and well thought out.

Component 2-3.

Page 7.

It seems to me that these components consist of both developing and preparing different tools, handbooks and action plans and an umbrella review together with concrete implementation strategies such as training and support which make this part a bit confusing. I recommend that you structure the section with project procedures in a table, see for example Seismann-Petersen et al. 2022: https://bmcnurs.biomedcentral.com/articles/10.1186/s12912-022-00858-6

If you find the suggestion with a table for the more detailed information you could describe the workflow in the components more briefly.

Component 4

As far as I understand, this component is a prerequisite for component 3 why I suggest that this would benefit from more logically being presented before the actual actions that will take place in component 3.

Component 5.

Page 9, 5.1 You describe that you will develop a core outcomes set for patient safety in perioperative care. Is it patient safety or quality of care or both. These concepts are often intermingled in the literature, I just want to draw your attention to it.

Page 9, 5.2 You are going to review health records of the participating hospitals and/or collect additional data to assess a) adherence to the standardised practices and b) surgical complications measured with the ISO tool. What exactly do you mean by reviewing health records is it systematic retrospective record reviews and on what level of data (patient data), please clarify.

The complications in the ISOS tool, are you going to use them as markers to be reviewed, could be clarified? What is the focus in the ISOS tool on complications, patient injuries or adverse events, needs to be clear?

How will you collect data according to adherence to standardized practices?

Page 9, 5.4 You write, each hospital will also be able to collect process and clinical data into the SAFEST platform, but what kind of data?

Survey of PROMs and PREMs, are you also going to collect a baseline for these variables?

Page 10, 5.5 Again, you could add references and motivate triangulation of data and sequential analysis. Is it a convergent parallel design you aim to do (see comment under method section)?

Component 6

Very good that you also intend to involve patients which is very important for both implementation and safe care.

Consider adding references on co-production/co-creation as I would say you aim to do.

Sustainability

There is a great strength that you intend to consider sustainability which the same as patient involvement is very important to achieve a sustainable project over time.

Analysis

The paper is lacking a clear and logic description of the analytic approach as it is embedded in the six main components with its sub-components or is it activities? If you do not provide a table is it possible to clarify this by a short summary or maybe a table specific for this section?

Discussion

This section seems very short to me as there is no discussion about design, and what the study will provide and what gaps that are addressed and according to what needs and if and what this project could act as a model for, but I will leave this to the editor to decide.

Data sharing statement

How will you handle patient record data included in the record review?

7. PLOS authors have the option to publish the peer review history of their article (what does this mean?). If published, this will include your full peer review and any attached files.

Reviewer #1: **Yes: **Prof Guy Haller

Reviewer #2: No

---

## [Author Response · Author response to Decision Letter 0]

30 Apr 2024

Dear PLOS ONE Editorial Staff and Reviewers;

We would like to express our sincere gratitude for taking the time to review our manuscript titled “Improving quality and patient safety in surgical care through standardisation and harmonisation of perioperative care (SAFEST project): a research protocol for a mixed methods study”. 

The reviewers’ insightful feedback and constructive comments have been invaluable in refining our work. In response to the reviewers’ suggestions and queries, we have prepared a detailed point-by-point response document named ‘Response to Reviewers’ (also copied and pasted below). Each point has been carefully considered, and we have provided explanations, clarifications, and revisions as appropriate. 

We are confident that this input has strengthened the quality and clarity of our manuscript, and we greatly appreciate your dedication to improving the work in our field.

Thank you once again for your valuable contributions.

Reviewer #1

General comments

This protocol describes a mixed-methods hybrid type III implementation study designed to support the development and adoption of evidence-based practices in order to improve perioperative patient safety. The study will be conducted in 10 hospitals of 5 European countries. Implementation outcomes include 1)adherence to standardised practices, 2)sustainability 3) usability, and 4) validity. The study will also assess adverse outcomes following surgery. The study protocol is well developed and the methodology adapted to the study purpose. 

Specific comments

Comment 1

The only point that should be clarified is the link between context analysis through the CIFER protocol and the specific intervention that will be implemented. The study protocol makes the assumptions that it will be a teaching program, but actually, the results of context analysis may suggest alternative strategies than teaching (i.e. computer program, change of the process of care, walk rounds etc…) This should be discussed in the protocol further. 

Response to comment 1

The Collaborative will serve as the main implementation strategy (see ‘Implementation strategies’ subsection under ‘Methods’), but indeed, it is expected that multiple implementation strategies that can impact the implementation will be employed at the sites. We have clarified that we will analyse this in section ‘4.4 Contextual factors evaluation’ by adding the following text:

“Data analyses will focus on how contextual factors serve as barriers and drivers to implementation and on the importance of the QILC in combination with other implementation strategies employed at the 10 hospitals.”

Comment 2 

There is also an imprecision in the introduction with the citation of the study by “Van Klei WA, Hoff RG, Van Aarnhem EE, et al. Effects of the introduction of the WHO "Surgical Safety Checklist" on in-hospital mortality: a cohort study. Ann Surg 2012;255(1):44-9 , as a reference showing that guidelines decrease surgical mortality. The study by Aarnhem is about a checklist, not about a guideline. I would suggest another reference. 

Response to comment 2

Thanks for the observation. We have included two references which are more appropriate for the statement below. 

“Adopting evidence-based practices can improve the quality of care delivered to the patients and thus significantly improve the overall safety outcomes of surgical care (Connor L 2023; Emond YEJJM 2022).”

New references:

1. Connor L, Dean J, McNett M, Tydings DM, Shrout A, Gorsuch PF, Hole A, Moore L, Brown R, Melnyk BM, Gallagher-Ford L. Evidence-based practice improves patient outcomes and healthcare system return on investment: Findings from a scoping review. Worldviews Evid Based Nurs. 2023 Feb;20(1):6-15. doi: 10.1111/wvn.12621. Epub 2023 Feb 8. PMID: 36751881.

2. Emond YEJJM, Calsbeek H, Peters YAS, Bloo GJA, Teerenstra S, Westert GP, Damen J, Wollersheim HC, Wolff AP. Increased adherence to perioperative safety guidelines associated with improved patient safety outcomes: a stepped-wedge, cluster-randomised multicentre trial. Br J Anaesth. 2022 Mar;128(3):562-573. doi: 10.1016/j.bja.2021.12.019. Epub 2022 Jan 15. PMID: 35039174.

Comment 3 

The limitation section is a bit vague since obstacles of implementation studies are well known and deserve attention (i.e. resources, training, duration). This should be specified in the limitation section.

Response to comment 3

We appreciate the feedback regarding the limitation section of our study protocol, we edited the second bullet point and added a third limitation (last bullet point) as follows:

Limitations

· There will be variability in interpreting how to implement and assess the level of adherence of the evidence-based practices; specific implementation and evaluation manuals will be developed to favour a similar understanding across participating hospitals.

· Various implementation challenges are anticipated in the application of the evidence-based practices, including potential resource constraints unique to each participating health facility. These challenges will be mitigated by developing context-specific implementation strategies.

· The study operates within a defined timeframe, limiting the ability to capture long-term outcomes and trends. Nevertheless, we've strategically allocated resources to ensure that the impact of our project extends well beyond the immediate study duration.

Comment 4 

In conclusion, a nice protocol of a promising study that needs however additional details provided.

Reviewer #2

General comments to authors

Thank you for allowing me to review the protocol for this ambitious but important project funded by the EU; Horizon Europe Framework Programme.

Since this is a substantial project I understand that it is a challenge to describe this in a balanced way in a protocol. But the paper will benefit from being restructured and clarified which will make it easier for the reader to grasp the project, see comments below. I think the main problem with the structure is that you have three processes (phases) sometimes running in parallel, that is the intervention (with planning and developing), the implementation and the evaluation process, but I have written some suggestions for this.

Although guidelines do not solve the whole issue to create safe care, to me this is an important prerequisite with impact on other initiatives that has to be solved. From a context perspective the need is extensive as the surgical specialty is responsible for many patient injuries or adverse events. I am looking forward to see future results from this project.

Specific comments

Comment 1

Title

Improving quality and patient safety in surgical care through standardisation and harmonisation of perioperative care (SAFEST project): a research protocol for a mixed methods study.

1. The title includes both concepts patient safety and quality but is not explained in the background why the reader get unsure how it is used in the different sections in the paper.

Response to comment 1

• We have now added the concept of quality into the Abstract and Background. In the Background we explained it as follows:

“Adopting evidence-based practices can enhance the quality of care delivered to the patients and thus significantly improve the overall safety outcomes of surgical care.”

Comment 2

Abstract section

2. Needs reformulation according to recommendations below.

Response to comment 2

• We edited the ‘Aim’ sub-section in the ‘Abstract’ as follows:

“SAFEST project aims to enhance perioperative care quality and patient safety by establishing and implementing widely supported evidence-based perioperative patient safety practices to reduce surgical adverse events.”

Comments 3-6

Background-Page 3.

3. The background is somewhat short but logic except from that some main concepts are missing (see further down) and there is no clear rationale (please add this).

4. Is it quality of care or patient safety that is your main focus? Please clarify.

5. Furthermore, your main concepts of quality of care or patient safety which also is your outcome measures are barely not mentioned nor defined in this section.

6. In the first section important information about adverse events is described. However, there is no clear definition of one of your main concepts i.e. adverse event (AEs). I find it very important that you define and are clear with what AE and patient injuries means for this project.

Responses to comments 3-6

• The main focus of this project is to improve patients’ safety throughout the surgical care pathway by enhancing the quality of care delivered through the implementation of evidence-based standardized practices. For this reason, the paper often uses the terms quality and safety together since by increasing the quality of care, patients’ safety will also improve. We edited the background section to better clarify these concepts. Additionally, we also define adverse events as suggested.

• Additionally, we edited the ‘Objectives’ section to better clarify the concept of quality of care in relation to patient safety.

Comments 7-11

Background-Page 4

7. Your objective is to improve and harmonise perioperative patient safety, but what does this mean. The focus in the background section is on unsafe care, clarify what you mean by perioperative patient safety? What do you want to achieve, is it less numbers of AEs which is implied? What about outcomes on a patient level? This needs to be developed.

8. Is it quality of care or patient safety or both that is your main focus? Please clarify.

9. One of the most important measures to achieve patient safety (safe care) in complex systems is to adopt a system perspective. According to me you are adopting a system perspective (system safety) in this project, which is very good. But the paper would benefit from a clearer description of this standpoint.

10. In the last section you write multidisciplinary teams; including experts on patient safety and perioperative care. However, this is the only place in your paper where they are mentioned. Since patient safety is your overarching objective, I find it very important that they will be included during the whole process.

11. Another concern is that you need to clarify what professions these multidisciplinary teams will consist of?

Response to comment 7

• To better clarify the project’s main objective, we edited as follows:

The SAFEST project, that will be conducted from June 2022 to June 2026, aims to improve and harmonise perioperative quality of care and patient safety by reducing surgical adverse events through the establishment and implementation of widely supported perioperative patient-centred standardised practices.

Response to comment 8

• As described above, the main focus of this project is to improve patients’ safety throughout the surgical care pathway by enhancing the quality of care delivered through the implementation of evidence-based standardized practices. For this reason, the paper often uses the terms quality and safety together as by increasing the quality of care, patients’ safety will also improve. We edited the ‘Background’ and ‘Objectives’ sections to better clarify these concepts.

Responses to comment 9

• As correctly mentioned by the reviewer, in order to increase the adherence to patients' safety practices, a system perspective is essential. For this reason, we will identify and analyse contextual factors (at macro-, meso- and micro-levels) that inhibit or promote the adoption, implementation, and sustainment of evidence-based patient safety practices for each hospital and related healthcare facility. This information will be used to develop recommendations on the implementation of the standards in hospitals in various contexts across Europe to reduce knowledge-practice gaps. Such rationale is explained in the ‘Objectives’ section of the manuscript as follows:

The SAFEST project, that will be conducted from June 2022 to June 2026, aims to improve and harmonise perioperative quality of care and patient safety by reducing surgical adverse events through the establishment and implementation of widely supported perioperative patient-centred standardised practices.

We also aim to increase the adherence to patients' safety practices by identifying contextual factors (at macro-, meso- and micro-levels) that inhibit or promote the adoption, implementation, and sustainment of evidence-based patient safety practices. This information may be used to develop recommendations on the implementation of the standards in hospitals in various contexts across Europe to reduce knowledge-practice gaps.

• In addition, we have edited a relevant paragraph of the ‘Background’ section to better clarify this concept as follows:

Therefore, having a set of patient-centred, evidence-based, practices with a high level of consensus and a comprehensive tailored implementation strategy considering contextual factors at macro-, meso- and micro-levels, is a cornerstone to ensure access to safe, innovative, sustainable, and high-quality surgical care.

Responses to comments 10,11

• We would like to clarify that the multidisciplinary teams mentioned in the ‘Background’ refer to the people who developed the SAFEST project, whereas the multidisciplinary teams mentioned in the rest of the manuscript starting from the ‘Design’ section refers to the members involved in the hospitals and related healthcare facilities in which the study will be carried out. To better clarify this, we have moved the ‘Setting’ sub-section at the beginning of the ‘Methods’ section and edited the text of the ‘Design’ subsection by adding a clear and detailed description including the roles of the multidisciplinary teams in the design section as follows:

The SAFEST study is a mixed-methods hybrid type III implementation study supporting the development and implementation of evidence-based patient safety practices through a Quality Improvement Learning Collaborative (QILC) strategy and combining qualitative and quantitative evidence through a convergent segregated approach. The QILC will adopt a participatory design involving multidisciplinary teams of patients and clinical care providers per participating hospital and related healthcare facility, including surgeons, anaesthesiologists, nurses, quality experts, IT representatives and primary care providers among others. The QILC teams from various hospitals will gather for learning sessions to share and adopt best practices towards a common improvement goal.

Comment 12

Objectives

12. To clarify this project, it would benefit from identifying overarching research questions. Since you have three main outcomes (clinical effectiveness, patients’ perspectives, and the process of implementation) in focus, this would be feasible.

Response to comment 12

• We have edited the ‘Objectives’ section to better clarify the aims of the study as follows:

The SAFEST project, that will be conducted from June 2022 to June 2026, aims to improve and harmonise perioperative quality of care and patient safety by reducing surgical adverse events through the establishment and implementation of widely supported perioperative patient-centred standardised practices.

We also aim to increase the adherence to patients' safety practices by identifying contextual factors (at macro-, meso- and micro-levels) that inhibit or promote the adoption, implementation, and sustainment of evidence-based patient safety practices. This information may be used to develop recommendations on the implementation of the standards in hospitals in various contexts across Europe to reduce knowledge-practice gaps.

• Additionally, we edited the ‘Outcomes’ section by including the following four research questions:

1. Research question: How effectively are patient safety practices being adhered to in perioperative care?

2. Research question: What is the impact of adherence to patient safety practices on clinical effectiveness?

3. Research question: How do patients perceive the safety and quality of their surgical care?

4. Research question: What are the key implementation process outcomes and how do these factors influence the successful implementation of standardized practices in surgical settings?

Comments 13-19

Methods- P

---

## [Editor Report · Decision Letter 1]

8 May 2024

Improving quality and patient safety in surgical care through standardisation and harmonisation of perioperative care (SAFEST project): a research protocol for a mixed methods study.

PONE-D-23-31626R1

Dear Dr. Valli,

We’re pleased to inform you that your manuscript has been judged scientifically suitable for publication and will be formally accepted for publication once it meets all outstanding technical requirements.

Kind regards,

Ann-Sofie Sundqvist, Ass. Prof.

Academic Editor

PLOS ONE
---

## [Editor Report · Acceptance letter]

21 May 2024

PONE-D-23-31626R1 

PLOS ONE

Dear Dr. Valli, 

I'm pleased to inform you that your manuscript has been deemed suitable for publication in PLOS ONE. Congratulations! Your manuscript is now being handed over to our production team.

Kind regards, 

on behalf of

Mrs. Ann-Sofie Sundqvist 

Academic Editor

PLOS ONE